# Reconstructing contact network structure and cross-immunity patterns from multiple infection histories

**Christian Selinger** *, **Samuel Alizon**

MIVEGEC, Univ. Montpellier, CNRS, IRD, Montpellier, France

* christian.selinger@ird.fr

**Data Availability Statement:** Code is accessible via zenodo repository: https://doi.org/10.5281/zenodo.5159448.

**Funding:** This work has been funded by the European Research Council (ERC) under the

## Abstract

Interactions within a population shape the spread of infectious diseases but contact patterns between individuals are difficult to access. We hypothesised that key properties of these patterns can be inferred from multiple infection data in longitudinal follow-ups. We developed a simulator for epidemics with multiple infections on networks and analysed the resulting individual infection time series by introducing similarity metrics between hosts based on their multiple infection histories. We find that, depending on infection multiplicity and network sampling, multiple infection summary statistics can recover network properties such as degree distribution. Furthermore, we show that by mining simulation outputs for multiple infection patterns, one can detect immunological interference between pathogens (i.e. the fact that past infections in a host condition future probability of infection). The combination of individual-based simulations and analysis of multiple infection histories opens promising perspectives to infer and validate transmission networks and immunological interference for infectious diseases from longitudinal cohort data.

## Author summary

Infectious disease dynamics are constrained both by between-host contacts and pathogen interactions within a host. The circulation of multiple parasites within a population constitutes a unique signature for each host's infection lifespan. We hypothesise that such individual multiple infection histories can inform us about the host contact networks on which parasites can be transmitted, but also on within-host interactions, where prior infections shape susceptibility to new infections. For proof-of-concept, we develop a simulator for multiple infections on networks. By combining intuitive novel metrics for multiple infection events and established tools from computational data analysis, we show that similarity in infection history between two hosts correlates with their proximity in the contact network. By analysing pathogens co-occurrence patterns within hosts, we also recover between-pathogen interference at the population level. The demonstrated robustness of our results in terms of observability, network clustering, and pathogen diversity opens new perspectives to extract host contact and between-pathogen interference information from longitudinal infection data.

European Union's Horizon 2020 research and innovation program (EVOLPROOF, grant agreement No 648963 to SA). The funders had no role in study design, data collection and analysis, decision to publish, or preparation of the manuscript.

**Competing interests:** The authors have declared that no competing interests exist.

## Introduction

Host populations are often assumed to be 'well-mixed' even though individuals tend to only interact with a small subset of the whole population and these contact patterns between individuals can dramatically affect the way epidemics spread [1–3]. For instance, it was shown during the early phase of the HIV pandemics that not only the average number of sexual partners but also the variance in the number of partners both increase the basic reproduction number ($R_0$) of sexually transmitted infections [4]. Since then, studies have identified how key parameters of the host contact network affect the risk of outbreak [5, 6] and the spread of an epidemic [7–13].

Network reconstruction, *i.e.* the inference of adjacency weights based on observations of a dynamical system acting on the network, is a well-established research topic in engineering [14], and has recently been applied to infectious disease dynamics [15–17]. In the field of epidemiology, measuring contact networks is a lively research topic [18], ranging from definition issues [19] (defining a 'contact'), to assessing the appropriateness of various types of data [20]. For livestock [21], interactions between interconnected farms can be well approximated through shipping logs. Other settings, such as wild populations or human populations, are more challenging to analyse. For humans, the field has traditionally relied on self-reported data, but new insights have been provided by airline transportation registries [22] or cell phone data [23]. More recently, parasite sequence data was used to infer network properties by analysing phylogenies of infections [24]. Importantly, this genetic data, which by definition pre-dates new outbreaks, can be used to make relevant predictions [25]. Note that we use the terms parasite and pathogen interchangeably encompassing both micro- and macro-parasites.

We hypothesise that host contact network properties can be inferred from individual longitudinal data about infection status (Fig 1). Such longitudinal data is classically used in epidemiological studies to measure the odds that a specific event may occur [26], however, they are rarely coupled to mathematical models of disease spread (but see [27, 28]). Our idea is that multiple infection status can be used as a unique descriptor of a host's position within the network. For instance, a host connected to many other hosts is expected to be more infected than a host connected to only one other host. Furthermore, hosts close in the network are expected to be infected at closer dates than hosts far away in the network.

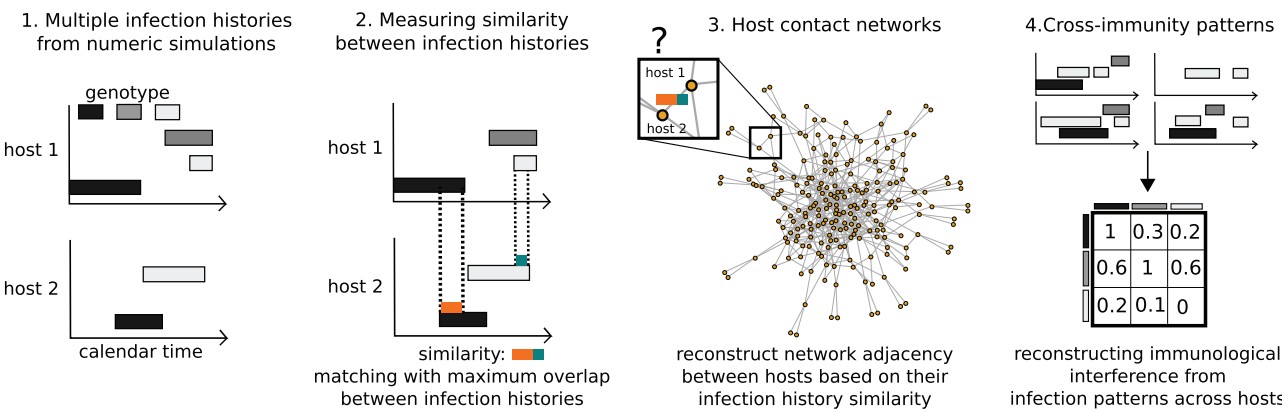

**Fig 1. Diagram illustrating the workflow for reconstruction of network properties and immunological interference.** In the first step, we obtain multiple infections histories of individual hosts from our simulator. Second, we calculate similarity metrics between hosts based on their multiple infection histories. Third, by comparing with simulated networks, we determine statistical associations between infection history similarity and contact network adjacency. Fourth, by calculating occurrences of infection overlaps across multiple infection histories, we obtain a matrix of immunological interference between pathogens.

To track multiple parasite strains or species in these individual histories is both feasible and highly relevant, as the increasing affordability and power of sequencing strengthen the assertion that most infections are genetically diverse [29]. In fact, in the case of genital infections by human papillomaviruses (HPVs), not only do we know that coinfections, i.e. the simultaneous infection by multiple genotypes, are the rule rather than the exception, we also know that they strongly correlate with the number of lifetime sexual partners [30]. Another classical example is the distribution of the number of macroparasites per host, which has been used to infer population structure [31]. This makes multiple infections an ideal candidate to measure epidemic properties but also detect potential within-host interactions between parasite strains or species [32, 33].

Practically, this simulation study is intended to mirror analyses that could be performed using field data. This could be based on PCR-based detection tests for different genotypes of a parasite species. These would then be performed in the context of longitudinal follow-ups of seasonal (e.g. [34] for respiratory viruses) or prevalent and long-lasting infections (e.g. [35] for HPV). Importantly, the inference could also be based on longitudinal serological data (i.e. information on past infection). This would be particularly adapted in the case of short infections (e.g. influenza) or infections that are likely to be asymptomatic. Overall, we hypothesise that such studies can inform us both on the contact network on which parasite species (or strains) are spreading, but also on their immunological interactions. The latter can be of interest between genotypes of the same species (e.g. influenza variants) but also between different species (e.g. cross-immunity between zika and dengue viruses [36]).

To show how individual infection histories for multiple parasite strains can inform us about the underlying transmission contact network, we conduct a simulation study, which requires alleviating two obstacles. First, we need to simulate multiple infections on a network, a task few studies have attempted [12, 37–42]. For this, we take advantage of recent developments in stochastic epidemiological modelling and implement a non-Markovian version of the well-known Gillespie algorithm [43]. This allows us to make sure a host's infection history is not lost every time it acquires a new strain. It also addresses statistical evidence that for many parasites infection duration does not follow exponential but heavy-tailed distributions [44–49]. The second main obstacle resides in extracting information from longitudinal follow-up data. To compare these infection histories, we use barcode theory inspired by computational topology [50], where sets of intervals -in our case infection onset and clearance- are compared between each other.

We integrate infection histories into our multiple infection modelling framework by accounting for the documented fact that recovery from infection becomes more likely with increasing 'age' of infection [51–54]. Then, we simulate individual multi-strain infection histories for epidemiological models with genotype-specific immunity on random clustered networks to demonstrate the impact of network topology on infection diversity. We refer to 'genotypes' to describe the parasite diversity, but our results apply to different genotypes of the same species or of different species of parasites provided that their mode of spreading (e.g. airborne, sexually transmitted) on the host network is the same. We include immunological interference between genotypes in our model by constraining the probability of acquiring a new infection in terms of the host's infection history, akin to hemagglutination inhibition assays (e.g. multi-season influenza strains [55]).

Finally, we show that infection barcodes can inform us both about the connectivity in the network and immunological interference between genotypes in several ways. First, similarity matrices between individual infection histories highly correlate with the network adjacency matrices. Second, an individual host's network connectivity can be inferred from its connectivity in terms of infection barcodes. Third, by quantifying genotype co-occurrence based on

infection histories, we recover model inputs for immunological interference. Fourth, mining frequent infection sequence motifs allows quantifying patterns induced by immunity settings in the spirit of more widely used cross-sectional prevalence surveys.

Taken together, in our simulation study we provide proof-of-concept to reconstruct network and immunity characteristics from novel summary statistics based on multiple infection histories. The demonstrated robustness and limitation of our approach towards network properties, host population sampling and genotyping opens novel avenues towards computational epidemiology of multiple infections.

## Materials and methods

### Simulation algorithm

We developed an event-driven stochastic model of multiple infections on networks in Python 3.7. For the purpose of this simulation study, we considered static, random networks to model contact (i.e. edges with binary weights) between hosts (i.e. nodes). Contact networks were generated using the class of random clustered graphs [56–58] implemented in the `networkx` package in Python [59]. Given a propensity list of edge degrees and triangle degrees as input, this algorithm generates a random graph with predefined average degree and average clustering coefficient. The clustering coefficient of a graph was defined as the local clustering coefficient (i.e. the degree of a node divided by the number of all possible edges in the node's neighbourhood) averaged over all nodes. Degree dispersion was defined as the ratio of degree variance and degree mean. Degree assortativity (i.e. the propensity for nodes in the network that have many connections to be connected to other nodes with many connections) was defined following [60]. For comparison purposes, using the same Python package, we also created random regular graphs with fixed degree of four. By definition, these graphs have zero degree dispersion.

Upon network initialisation, we randomly seeded outbreaks of one infection per genotype simultaneously, allowing for up to four genotypes per host at high multiplicity. Disease dynamics followed Gillespie's stochastic simulation algorithm (SSA) [61, 62] adapted to a non-Markovian setting [63] to incorporate memory-dependent processes. Here, in particular, we considered recovery from infection as a process depending on the age of infection.

The simulations also feature potential immunological interference between genotypes. This was defined by an immunity input matrix $J_{ij} \in [0, 1]$, where $J_{ij}$ is the probability not to acquire an infection with genotype $i$ given prior infection with genotype $j$, i.e. upon exposure from an infectious edge with genotype $j$. We sampled independent Bernoulli random variables with probabilities $J_{ij}$, for all genotypes $i$ contained in the infection history of the exposed host. In these simulations, we explored five immunity settings defined as follows:

$$J_1 = \begin{bmatrix} 1 & 1 & 0 & 0 \\ 1 & 1 & 0 & 0 \\ 0 & 0 & 0 & 0 \\ 0 & 0 & 0 & 0 \end{bmatrix} \quad J_2 = \begin{bmatrix} 1 & 0 & 0 & 0 \\ 0 & 0.8 & 0 & 0 \\ 0 & 0 & 0.6 & 0 \\ 0 & 0 & 0 & 0.4 \end{bmatrix} \quad J_3 = \begin{bmatrix} 1 & 0 & 0 & 0 \\ 0 & 1 & 0 & 0 \\ 0 & 0 & 1 & 0 \\ 0 & 0 & 0 & 1 \end{bmatrix}$$

$$J_4 = \begin{bmatrix} 1 & 0.8 & 0.8 & 0.8 \\ 0.2 & 1 & 0.8 & 0.8 \\ 0.2 & 0.2 & 1 & 0.8 \\ 0.2 & 0.2 & 0.2 & 1 \end{bmatrix} \quad J_5 = \begin{bmatrix} 1 & 1 & 1 & 1 \\ 1 & 1 & 1 & 1 \\ 1 & 1 & 1 & 1 \\ 1 & 1 & 1 & 1 \end{bmatrix}$$

The matrix $J_1$ corresponds to partial cross-immunity, with sterilising immunity between the first two genotypes, $J_2$ models decreasing homologous immunity, where genotype $g_1$ induces sterilizing immunity, whereas immunity is decreasing for the remaining genotypes. $J_3$ models homologous immunity, i.e. when each genotype induces sterilizing immunity to future infection with the same genotype. $J_4$ is an example for asymmetric cross-immunity, i.e. immunity to infection with $g_1$ after clearing an infection with $g_2$ is stronger than immunity to infection with $g_2$ after clearing an infection with $g_1$. Finally, $J_5$ models sterilizing cross-immunity. Transmission and recovery rates were considered equal for all genotypes and fixed in advance.

The underlying parasite life-cycle in our epidemiological models can be interpreted as a variation of a classical SIRS model, where susceptible hosts ($S$) can be infected upon contact with an infected host ($I$). Infected hosts become immune to infection after recovering ($R$) but this immunity can wane making the hosts susceptible again ($S$). In absence of host immunity or in presence of lifelong immunity, the life-cycle converges towards that of an SIS or an SIR model. However, there are two important differences with these canonical models [64]. First, our model allows for multiple infections meaning that host immunity to a given genotype depends on past infections (i.e. the $J$ matrixes). Second, our model is non-Markovian meaning that the probability that an event occurs (e.g. recovering from an infection) depends on the history of the infection (i.e. the number of days elapsed since the inoculation).

The Gillespie SSA allowed us to simulate these disease dynamics by performing regular updates in the values of the rates. For this, at each time point, we first created a rate vector $\{r_k\}_{k \in E}$ indexed by the set of all possible events $E$ (recovery from an ongoing infection or acquisition of a new infection from an infectious host in the graph neighbourhood). If the node $i$ had spent $t_{i,g}$ time in an infection with genotype $g$, then the rate of recovery from this infection was assumed to be given by a Weibull hazard rate function $h(t_{i,g}, \alpha_g, \mu) = \alpha_g \mu_g t_{i,g}^{\alpha_g - 1}$ with shape parameter $\alpha_g$ and scale parameter $\mu_g$. The default setting for this recovery rate function was $\alpha_g = 2$ and $\mu_g = \frac{1}{\Gamma(1 + \alpha_g^{-1})}$ such that the mean was equal to one and the variance equal to 2.27, with $\Gamma$ being the well-known gamma function. The probability of clearance increased with the age of infection, motivated by infection duration literature (see Introduction). For normalisation purposes, we assumed that the rate of infection for any node currently not infected with genotype $g$ was unity, such that the average number of secondary infections was uniquely determined by the average node degree.

To better handle the computational workload of multiple infections on networks, we stored the infection histories in an interval tree data structure containing the times of onset and end of each infection episode, the genotype, and the host number (this is an individual-based model and not an event-based model). We refer to this data structure as infection barcodes (see below). At a given time, from the rates vector, we first drew an exponentially distributed random variate with rate $\sum_k r_k$ to determine the time increment to the next event. Certain rates $r_k$ depended on the infection age but, as shown in [63], drawing exponentially distributed random variables with respect to these rates provides good Markovian approximations of non-Markovian processes, granted the number of events is sufficiently large. We then drew a random variate according to the total probability vector $\{r_k / \sum_k r_k\}_{k \in E}$ to determine the nature of the next event. Depending on the event, we updated the list of possible events. In the case of recovery of host $i$ from infection with genotype $g$, we wrote the end time of the infection episode to the interval tree, removed the recovered infection from the rate vector, and added potential infections with genotypes other than $g$ from the host's network neighbourhood. In the case where a host $i$ is newly-infected with $g$, we wrote the time of onset of the infection episode to the interval tree, removed all possible edges from $g$-infected neighbours of $i$, and added possible infection edges from $i$ to all neighbours that had not been or were not currently

infected with $g$. We then updated the rates vector again and proceeded until the epidemic became extinct.

Unless stated otherwise, we simulated disease transmission with four genotypes introduced simultaneously on the giant component of random clustered graphs with 250 nodes with average degree of 4 and an average clustering coefficients of 0.34 (referred to as 'clustered' networks). For a given parameter set, we seeded a random outbreak at the beginning of each simulation with one infection per genotype and ran 50 stochastic replicates until the disease-free state or a pre-defined time horizon was reached. Epidemic outcomes were reported as average and 95% confidence intervals for equidistant time bins.

The source code and configuration files used for simulations are publicly available at the zenodo repository: https://doi.org/10.5281/zenodo.5159448.

## Genotype diversity and multiplicity of infection

We measured genotype diversity during the course of epidemics with $N = 4$ genotypes by Shannon's diversity index [65] defined as $H(t) = \exp(-\sum_i p_i(t) \log p_i(t))$, where $p_i(t)$ is the frequency of infections with genotype $i$ relative to all infections present at time $t$. The index $H \in [1, 4]$ is maximised when all genotypes have equal frequency and minimised when only one genotype is present, and we used the implementation of the Python module `ecopy`. We emphasise that this index pertains to diversity at the population-level and does not distinguish between a multiple infection within a single host and single infections in several hosts. To highlight the importance of multiple infections, we also considered the multiplicity of infection (MOI) given by the number of genotypes present within infected hosts at a given time. We reported MOI averaged over all infected nodes.

## Infection barcodes and network properties

We summarised each individual multiple infection history by a barcode, i.e. the set of intervals describing the onset and clearance of infection episodes accumulated within a host during the course of an epidemic. The notion of barcode first arose in computational topology [66] to succinctly summarise and compare topological properties of metric spaces. Mathematically, the infection barcode of a host $A$ is a set of triples $A = \{A_1, \ldots, A_n\}$, where each $A_i = (b_i, d_i, g_i)$ defines the birth $0 \le b_i < \infty$ (i.e. infection onset) and death $0 \le d_i \le \infty$ (i.e. infection clearance) with a particular genotype $g_i$. We explicitly included the presence of multiple genotypes in a host's infection history, i.e. a host can simultaneously have several infections, also the same genotype can appear several times in the course of an epidemic. To compare infection histories between hosts, we considered the metric space of infection barcodes endowed with two complementary notions of distance.

In the first approach, we considered a similarity index [67] such that two hosts with largely overlapping infection episodes tended to be similar to each other, regardless of the respective genotypes. More precisely, a given pair of barcodes between any two hosts $A = \{A_i\}$ and $B = \{B_j\}$ was transformed into a weighted bipartite graph. The nodes of this graph consisted of infection episodes and these nodes were partitioned according to the two hosts. The edge weight of the graph was given by the overlap length $|A_i \cap B_j| = |[b_i^A, d_i^A] \cap [b_j^B, d_j^B]|$ between the infection episodes constituting the edge $(i, j)$, taking also into account overlaps between distinct genotypes. To obtain the similarity index $s_1(A, B)$, we calculated the maximum bipartite graph matching of this graph. The similarity index was transformed into a distance by $d_1 = -2s_1 + \sum_i |A_i| + \sum_j |B_j|$.

For the second approach, we considered the overlap length of infection episodes between two hosts with matching genotype $g$. By summing over all genotypes we obtained the similarity

index $s_2(A, B) = \sum_g \sum_{ij} \mathbb{1}_{g_i = g_j = g} |[b_i^A, d_i^A] \cap [b_j^B, d_j^B]|$. This index was transformed into a distance by setting $d_2 = -2s_2 + \Sigma_i |A_i| + \Sigma_j |B_j|$.

To determine whether similarity between infection histories implies proximity in the transmission network, we compared the metric space of infection barcodes to the metric space of the network, restricted to nodes that had been infected during the course of the epidemic. For the network, we considered two different distance notions, i.e. the discrete metric based on the binary adjacency matrix and the shortest path distance of the graph. For the shortest path distance, we expected negative correlations with barcode similarity, since the further two nodes are apart from each other (i.e. the longer the shortest path distance), the less infection barcodes would be similar to each other. The converse holds for the adjacency, since adjacent nodes with unit weight would be similar in terms of infection histories.

We used two-sided p-values from the Mantel test [68] between the barcode similarity (resp. distance) matrix and the adjacency or shortest path length matrix respectively to assess for each model the correlation between infection barcode and network topology [69]. Since Mantel permutation tests have been scrutinised for underestimating type I errors in the presence of spatial autocorrelation [70–72], we tested adjacency and shortest path matrices for auto-correlation, using the R package statGraph [73]. While adjacency matrices were not significantly auto-correlated for most lags, the shortest path matrices had significant auto-correlations at all lags (Fig G in S1 Text).

To assess whether spatial correlations between similarity matrices also translated to local properties such as a host's node degree (i.e. number of network neighbours) or its localisation within the network (i.e. the sum of shortest path lengths to other hosts), we developed a measure of infection barcode connectivity and performed regression analyses of network characteristics on infection connectivity. More precisely, for a given simulation with maximum length $L$, $N_g$ different genotypes and $D + 1$ nodes in the giant component, the normalized infection barcode connectivity of a host $n$ was defined by

$$\hat{s}_i(n) = (LN_g D)^{-1} \sum_m s_i(n, m) \in [0, 1]$$

for $i = 1, 2$, i.e. the sum of all barcode similarity values within the network of infected hosts relative to the maximum possible barcode similarity. Since shortest path length is a continuous measure, we performed linear regression using infection barcode connectivity as a regressor and assessed significance for coefficients by p-values from two-sided Z-tests for each of the immunity and network clustering settings. The relationship between node degree and infection connectivity was assessed using a multinomial logistic regression model defined by

$$\log \frac{\mathbb{P}(\text{degree} = k)}{\mathbb{P}(\text{degree} = 1)} = \beta_0 + \beta_1 \hat{s}_i + \beta_2 N_g + \beta_3 c$$

where $\beta_0$ is the intercept, $\beta_1$ is the coefficient for continuous variable of the normalised barcode similarity degree, $\beta_2$ is the coefficient for the categorical variable of $N_g = 2$ relative to the base level $N_g = 4$, and $\beta_3$ is the coefficient for the categorical variable of clustering coefficient $c = 0.17$ relative to the base level $c = 0.34$. We evaluated the model for genotype-agnostic $\hat{s}_1$ and genotype-specific $\hat{s}_2$ similarity degrees for each of the immunity settings. The multinomial model was trained on 80% of the stochastic replicates and test against the remaining 20%, the area under the curve (AUC) from multinomial precision call curves [74] was used to evaluate predictive power with the R package pROC.

### From infection barcodes to immunity

In order to determine immunological interference between genotypes based on individual infection histories, we quantified non-overlapping co-occurrence $C(G, H)$ of any two genotypes $G$ and $H$ within the host population. Large $C$ values indicate that the presence of genotype $G$ in the infection history did not preclude subsequent infection with $H$, i.e. the immunological interference between the two genotypes was weak. Conversely, if $C$ is zero, then infection with $G$ did prevent infection with $H$, such that $G$ conferred full cross-immunity to $H$. This can be written mathematically as

$$C(G, H) := \sum_{A} \sum_{A_i, A_j \in A} \mathbb{1}_{\{b_i > d_j,\ g_i = G,\ g_j = H\}} \tag{1}$$

As an alternative, we used the sequence mining algorithm cSPADE [75] to determine the most frequent infection patterns from infection histories in the network. This algorithm has originally been designed to mine and classify the most frequent patterns in sets of sequences, e.g. products purchased by customers over multiple time points. In the machine learning literature, the frequency at which an item (e.g. a product, or a set of products) is encountered across a set of time-stamped customer data is referred to as 'support'. In order to determine the most frequent multiple infection patterns at various snapshots during an epidemic, we interpreted multiple infection data as such sequences. More precisely, we sampled five random observation times $\tau_1, \ldots, \tau_5$ during a simulation, and, for each host (i.e. 'customer'), we defined the sequence of infections (i.e. 'purchases') by the set of genotypes the host was infected with at each observation time point. We deliberately assumed for this approach that multiple infection sequence motifs were independent snapshots of the infection state, such that re-infections and persistent infections were indistinguishable. Since our simulator allowed for multiple infections, elements of a sequence comprised the empty set, singleton sets (with only one genotype), or sets of several genotypes. The length of a motif is equivalent to the number of observations (e.g. the motif of length two $< \{g_1, g_3\}, \{g_3\} >$ indicated that a double infection was observed within hosts at the first time point and that a single infection was observed at the second time point). We used the `R` package implementation `arules` of the cSPADE algorithm and calculated the frequency (i.e. the 'support') averaged across simulation replicates and sequence motifs with minimum support of 0.02 and maximal length of 5.

## Results

### Network topology, genotype diversity, and MOI

To better understand how the topology of a contact network impacts multiple infection dynamics, we simulated epidemics by introducing simultaneously infections on random clustered graphs with distinct summary statistics. By default, we simulated multiple infections on random clustered graphs with high clustering coefficient, which we refer to as 'clustered' networks (see Table 1). For comparison, we considered 'dispersed' random clustered graphs, emulating contact networks with relatively low but variable number of contacts spread out evenly in the network due to highly over-dispersed degree and low clustering coefficient. To determine whether these networks would result in contrasting epidemic dynamics, we also considered single genotype simulations resulting in larger final epidemic size for dispersed networks (Fig A in S1 Text). In order to uncover potential biases stemming from degree dispersion and clustering, we also simulated multiple infections on random regular graphs, which had fixed degree of four and clustering coefficient of 0.014.

**Table 1. Summary statistics of the two types of networks simulated.** Quantities are averaged across 250 nodes and 50 stochastic replicates.

| Network type | degree | degree assortativity | degree dispersion | clustering coefficient |
|---|---|---|---|---|
| dispersed | 4.6 | 0.58 | 51.51 | 0.13 |
| clustered | 4.1 | 0.53 | 2.36 | 0.34 |

Infection multiplicity (MOI) was highest for both homologous immunity settings, which also maintained a high diversity over the course of the simulated epidemic (Fig 2). In general, the network topology only had a marginal impact on MOI and diversity, with dispersed networks (i.e. high degree and low clustering coefficients) showing slightly higher diversity (e.g. for homologous decreasing and asymmetric settings).

Diversity decreased initially in all the runs, which was likely due to stochasticity (some strains grow more than others). The striking differences in diversity appeared later on with three groups with distinct shapes (Fig 2): (i) partial cross-immunity maintained moderate diversity throughout, (ii) homologous and homologous decreasing immunity had diversity following the fraction of infection nodes, with a sharp increase and decrease towards the end of the simulation, and (iii) asymmetric and full cross-immunity showed continuously declining diversity. As expected, full cross-immunity prevented multiple infections and sharply decreased diversity, especially for clustered networks. Infection diversity on random regular networks mirrored in general those on clustered networks, with a markedly lower levels for partial cross immunity settings (Fig B in S1 Text). Multiple infections could be sustained in the partial cross-immunity setting, under which the fraction of infected nodes showed a SIS-like shape due to reinfections by a subset of the circulating genotypes.

## Infection barcodes and network structure

To infer properties of the host contact network from individual infection histories, we summarised each of these through an infection barcode, i.e. a set containing all intervals of infection episodes with different parasite genotypes. In order to measure similarity (or distance) between infection barcodes of any two hosts, we used the length and frequency of overlaps between infection episodes. As detailed in the Materials and methods, we distinguished

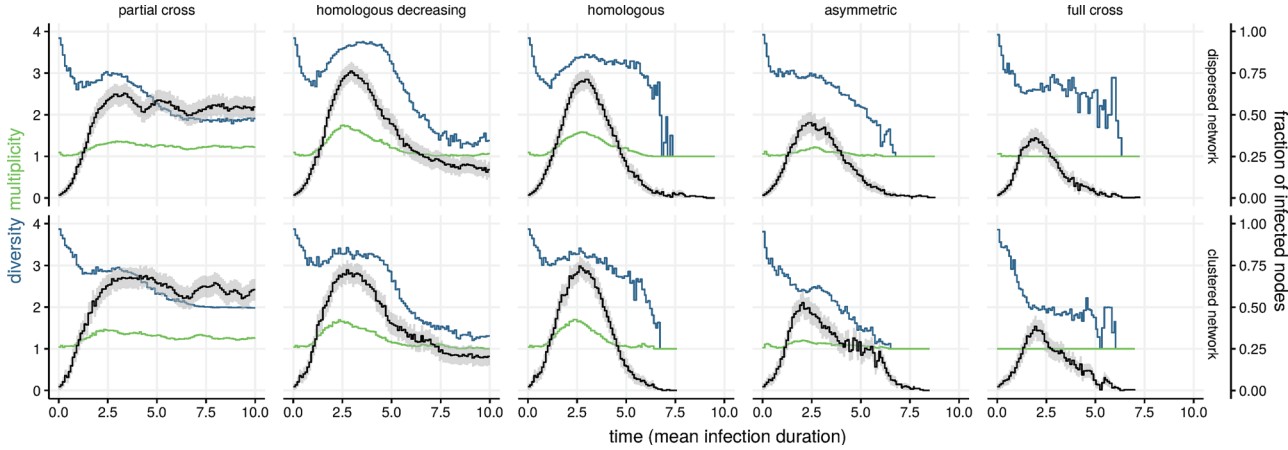

**Fig 2. Genotype diversity index, multiplicity of infection (MOI), and epidemic prevalence as a function of network type and cross-immunity patterns.** The figure shows the output of 50 stochastic epidemics with four genotypes on dispersed and clustered networks (see Table 1), and five immunological interference settings. Black lines show the time-averaged fraction of nodes infected with at least one genotype (95% confidence intervals shaded grey), blue lines show the average genotype diversity index at the population level, and green lines show the average MOI of individual nodes.

overlaps that were genotype-specific (i.e. only for matching genotypes between two hosts) from those that were genotype-agnostic (i.e. regardless of the genotype). The resulting barcode matrices quantified the similarity (or distance) between any two hosts. These were tested for correlation to more classical matrices that capture global (shortest path between nodes) or local properties (binary adjacency) of contact networks restricted to infected individuals.

Simulation results with four different parasite genotypes show that, if the network of infected individuals was fully observed, the barcode similarity matrix significantly correlated with neighbourhood properties of the contact network given by its adjacency matrix (Fig 3). In more realistic situations, however, infected nodes would only be partially observed. Our results showed that spatial correlations were less significant as the sampling rate of network nodes decreased but remained significant for sampling proportions of 25% or more.

When pairwise comparisons between hosts were only based on the number of genotypes (green bars) but not on their nature (blue bars), our ability to detect significant correlations decreased. Unsurprisingly given the importance of multiple infection histories for our approach, decreasing the number of circulating genotypes from four (dark bars) to two (light bars) also decreased correlation significance. Immunity assumptions allowing for multiple infections with highly diverse infection barcodes (e.g. partial cross, homologous decreasing, homologous) resulted in higher significance levels. Likewise, when tested for global network properties given by the shortest path matrix (bottom in Fig 3), significance levels and correlation values (Fig C in S1 Text) appeared to be strongest for the same set of immunity assumptions. Overall, correlations with shortest path matrices were consistently less robust towards multiplicity than those with adjacency matrices (especially for the full cross immunity setting). This suggests that in order to distinguish distances between hosts beyond their graph neighbourhood with barcode similarities, one needs sufficiently rich multiple infection histories. Compared to clustered networks, we observed similar trends on dispersed networks regarding the immunity setting (Fig D in S1 Text), and correlations tended to be stronger than in the clustered network setting, especially for comparisons of barcode similarity matrices with

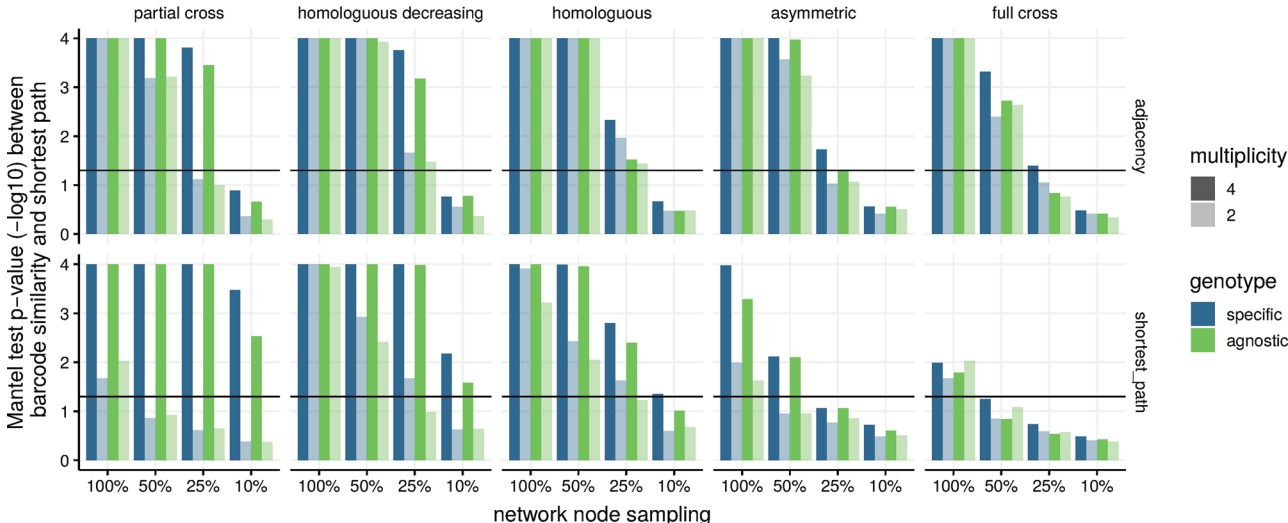

**Fig 3. Correlations between a matrix based on infection barcodes similarity and the network's adjacency (top) or shortest path (bottom) matrices.** For simulated epidemics on clustered networks with 2 or 4 circulating genotypes and a variety of cross-immunity settings, we tested for correlations using p-values of two-sided Mantel tests with $10^4$ permutations. For each setting, we re-sampled 20 times randomly 100, 50, 25, or 10% of the infected nodes and report the average p-value.

shortest path matrices. Overall, we found that the genotype-specific similarity index for infection barcodes was more informative than the distance metric (Fig E in S1 Text).

The spatial correlations between barcode similarity indices and both local and global graph properties also translated to the individual level. From the barcode similarity matrix, we calculated each host's connectivity (i.e. degree) in terms of infection history by summing the host's barcode similarity with respect to all other infected hosts and normalising appropriately (see Materials and methods). We hypothesised that the infection barcode connectivity within the network of infected hosts could inform the degree of connectivity in the contact network. First, the significant relationship between the barcode and shortest path connectivity indicated that a host with an infection history similar to those of many other hosts had also lower shortest paths to other hosts, and was hence more centrally located in the network (Fig 4).

Multinomial regression tests showed that the odds of having higher degree in the contact network (i.e. more neighbours) increased with infection barcode connectivity. The power of the model to predict node degree was dependent on the immunity settings and highest for partial cross-immunity (Fig 5), mirroring what was seen for the continuous measure of shortest path lengths (Fig 4). For this particular immunity setting, the area under the curve was 0.74, indicating that predictive power increased when using multiple infection data, especially because in this case diversity was maintained over the entire time horizon yielding highly specific barcode similarity scores. Similar trends, but with less predictive power, were observed for the remaining immunity settings.

## Infection history and immunological interference

To test whether immunological interference between genotypes could be inferred from individual infection histories, we simulated epidemics on random clustered networks with four distinct genotypes under a variety of immunity assumptions (see Materials and methods). The

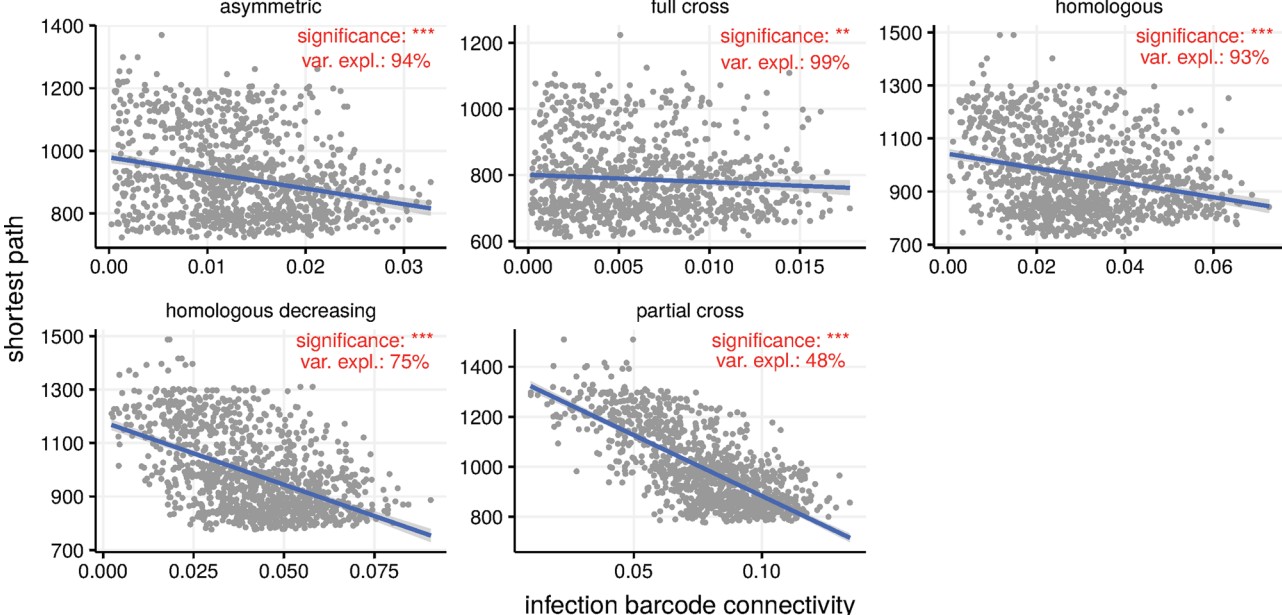

**Fig 4. Link between the host connectivity estimated via the barcodes and that estimated via the shortest path.** Hosts that are found to have a high connectivity according to the multiple infection history (the barcode metric) tend to be closer in the contact network. Significance levels corresponds to p-values of a linear regression ***: $p < 0.001$ and **: $p < 0.01$. The variance explained (in percent) by barcode connectivity refers to sum of squares from ANOVA analysis.

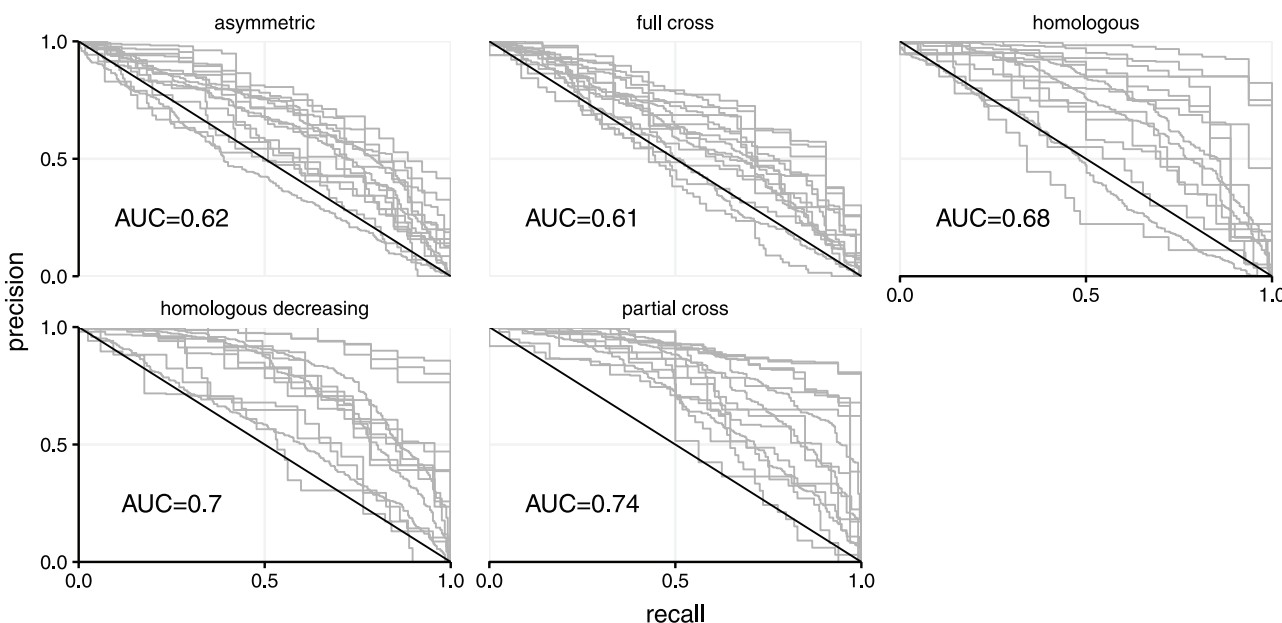

**Fig 5. Precision-recall curve for multinomial models under five distinct immunity assumptions.** Precision (resp. recall) is defined as the percentage of true positives among all positives (resp. true positives plus false negatives). The area under the curve (AUC) indicates increased (compared to random AUC of 0.5) power to classify a host's node degree using infection barcode connectivity information.

co-occurrence score $C$ between genotypes resembled the immunity input matrix from the simulations, in two ways. First, whenever immunity was sterile, we did not observe co-occurring genotypes (red squares in Fig 6). Second, when the probability to develop protective immunity following infection decreased, the co-occurrence score increased.

The partial cross-immunity setting allowed for co-occurrence between genotypes $g_3$ and $g_4$, while cross-immunity between $g_1$ and $g_2$ lowered also possible co-occurrence with genotypes $g_3$ and $g_4$. Similarly, decreasing homologous immunity strongly limited co-occurrence of the same genotype, and interestingly also limited co-occurrence between genotypes which were a priori not assumed to interfere immunologically (e.g. $g_1$ and $g_4$) due to population dynamics effects. The co-occurrence score was also able to capture asymmetric immunity, stressing the possibility to estimate the order of acquisition from infection barcodes.

The sequence motif mining approach yielded infection patterns that were consistent with the immunity input (Fig 7). For partial cross-immunity, multiple infections including genotypes $g_1$ and $g_2$ were absent from the motifs, whereas infections including the remaining genotypes were abundant with high frequency. The homologous decreasing setting was mirrored by a continuous increase in motifs from genotypes $g_1$ to $g_4$. In the homologous setting, motifs were dominated by single infections, whereas double infections were equally frequent between genotypes. Unsurprisingly, sterilising cross-immunity excluded all multiple infections. In order to distinguish the order of genotype acquisition, we had to consider motifs of length two (Fig F in S1 Text). In this case, differences in motif frequency with asymmetric immunity assumptions (e.g. the motif $< \{g_4\}, \{g_1\} >$ was more frequent than $< \{g_1\}, \{g_4\} >$) could be detected. In general, the higher computational efficiency of sequence mining compared to the co-occurrence score was offset by the fact that, in the former, motifs are independent snapshots of the infection state, which makes reinfections and persistent infections indistinguishable.

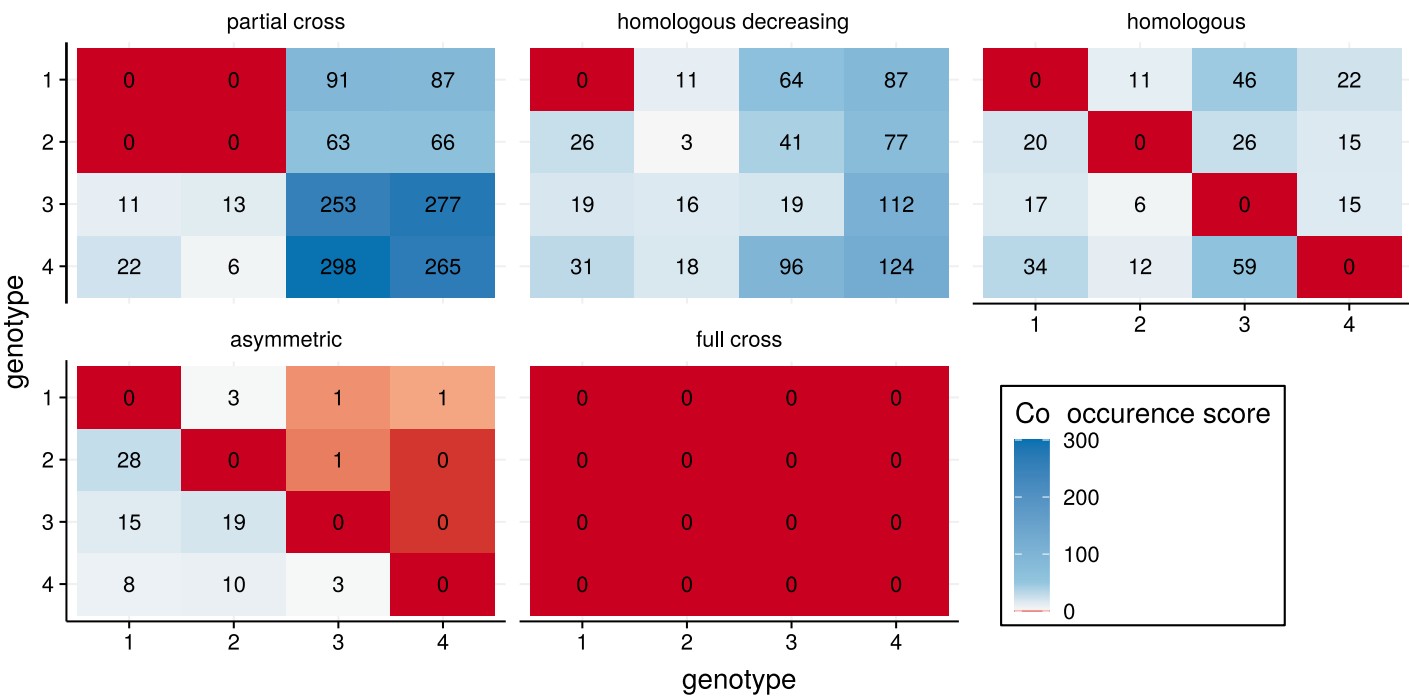

**Fig 6. Co-occurrence score from infection barcode outputs of simulations with various assumptions on immunological interference.** The underlying cross-immunity matrices are shown in the Materials and methods. The combinations with 100% cross-immunity correspond to the red cells with 0 co-occurrence score.

## Discussion

Understanding the properties of host contact networks is key to predicting epidemic spread but raises many practical challenges [18]. Phylodynamic studies have shown that some of these properties can be inferred from microbial sequence data [24, 76]. Here, we use individual infection histories to gain insights into the contact network structure.

The first obstacle was to simulate epidemics on contact networks while allowing for coinfections, i.e. the simultaneous infection of hosts by multiple parasites [77]. To enable clearance events based on a host's multiple infection history, we implemented a non-Markovian version of the Gillespie algorithm following recent developments in computational physics [43, 78]. As expected, network topology directly impacted multiple infection dynamics, with an increased level of clustering leading to higher parasite strain diversity.

Being able to simulate epidemics of multiple infections on networks, we sought to compare complex individual infection histories in order to reconstruct transmission networks. To address this issue, we captured these histories using barcodes and compared infection histories between hosts using tools from computational topology [50]. We could show that global properties of the network are correlated with these barcodes, i.e. similarity matrices inferred from the barcodes correlate with matrices inferred from the network adjacency matrix. Furthermore, individual-centred properties such as a host's degree can also be inferred from infection barcodes.

Detecting within-host interactions between pathogens from population-level data is relevant both for persistent (*e.g.* HPV [79, 80]) and acute (*e.g.* influenzavirus [81, 82]) infections. In this simulation study, we focused on possible immunological interference between pathogens leading to exclusion mechanisms for multiple infections. We show that some patterns can be identified using multiple infection data.

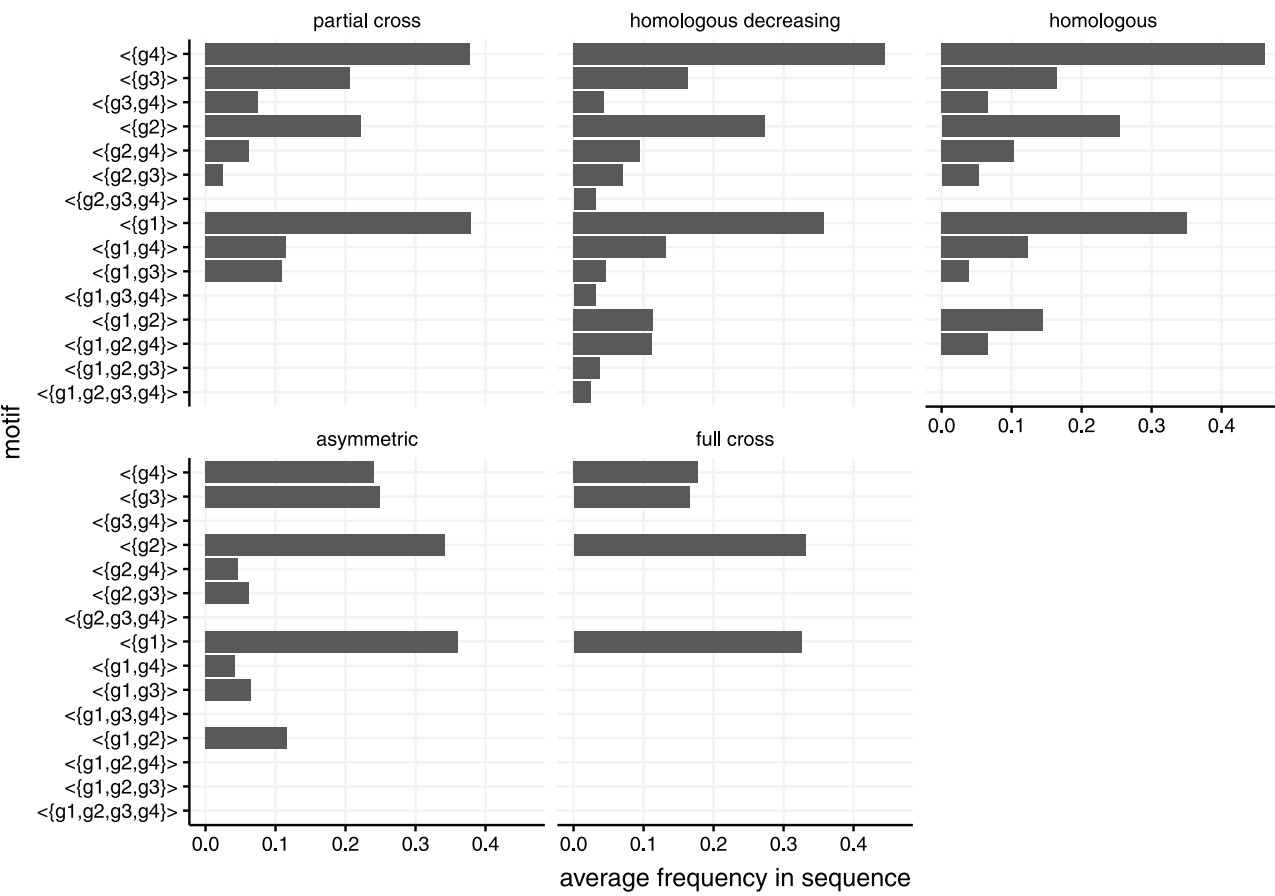

**Fig 7. Effect of cross-immunity patterns on the average frequency of all possible sequence motifs with length one.** The motifs represent cross-sectional multiple infection snapshots. Here, samplings are performed at five random time points during the simulation. Note that coinfection by all 4 genotypes is only found in the homologous decreasing case. The corresponding cross-immunity matrices are shown in the Materials and methods. See also Fig 6 for additional information regarding co-occurrence between the genotypes.

Overall, we provide proof-of-principle that multiple infections and infection history can be used to gain insight into host contact network properties and immunological interference. This is biologically sound, given that infections by multiple parasite genotypes are extremely frequent, and realistic. Indeed, for many human infections, there are longitudinal follow-ups with a detailed record of parasite diversity, one of the clearest examples being human papillomaviruses [83]. Studies also exist that follow individuals in natural populations [84, 85].

Individual-based models are particularly amenable to determine new ways to analyse multiple infection data from community-based routine surveillance cohorts [81, 82]. Our model is tailored towards quantities that could actually be observed in the field, e.g. the average number of contacts and the clustering coefficient can be extracted from surveys [86]. How precise genotyping should be is a highly relevant question. Furthermore, the time component is essential to distinguish ongoing infection from reinfection. However, for many acute infectious diseases, this issue can be addressed by spacing sampling time points.

There are several limitations in our work with possible extensions. First, we only reported results obtained on random clustered networks. We also obtained similar results on other types of topologies (e.g. random regular graphs) but it would be valuable to know whether infection history is more or less informative depending on the type of network considered and if network comparison can be performed. We also assumed that the contact network was

static, but in reality, it can be highly dynamic. Recent evidence suggests that these dynamics could be detected in sequence data [87] and it would be interesting to explore this with infection histories. Also, real-world multiple infection histories might be incomplete or noisy. Determining how sensitive statistical associations between barcode similarity and network properties are, could help delineate practical limits of our approach.

We assumed the life-cycle and transmission mechanisms of the parasites to be equivalent. Simulating parasite spread with distinct infectivity and clearance parameters could show the robustness of barcode metrics. Also, within-host dynamics such as viral load and immune memory could further identify relevant mechanisms for population-level dynamics of multiple infections. Following many existing studies, e.g. in phylodynamics, we used a neutrality assumption. However, genotypes are known to interact [88] and multiple infections can be a means to detect these interactions [79, 89]. In general, exploring richer life-cycles is a promising path for future studies.

We considered that infection history was based on parasite genotype presence in the host (e.g. PCR detection test) but the same methodology could be applied to serological data, i.e. evidence of past presence in the host via antibody detection. A clear advantage of this type of data is that it is more abundant. Another advantage is that it does not require a detailed longitudinal follow-up. However, the downside is that we then ignore the origin of the infection. Simulation studies could be instrumental in assessing our ability to infer network properties from serological data.

A separate line of future research has to do with the inference *per se*. One possibility could be to use Approximate Bayesian Computation to get a more precise idea of the accuracy of the prediction we can make on key network topology parameters. This could also allow us to compare between different classes of networks, e.g. using random forest algorithms [90].

Finally, with the increased power and decreasing cost of sequencing, it may be possible in the future to have both the information about infection history and the sequence data for multiple pathogens. It would then be worth determining whether we can get the best out of the two types of data, infection history being less precise but also less noisy than phylodynamic inference.

## Supporting information

**S1 Text.** Fig A. Final epidemic size (in percent) for single genotype infections on dispersed (red) and clustered (blue) networks. Fig B. Genotype diversity index, multiplicity of infection (MOI), and epidemic prevalence for random regular graphs for five cross-immunity patterns. Fig c. Spatial correlations for clustered networks for five cross-immunity patterns. Fig D. Spatial correlations for dispersed networks for five cross-immunity patterns. Fig E. Correlation tests between a matrix based on infection barcodes distance and the network's adjacency (top) or shortest path (bottom) matrices. Fig F. Mining for motifs of length two. Fig G. Spatial auto-correlations for dispersed (left) and clustered (right) neworks.
(PDF)

## Acknowledgments

The authors acknowledge additional support from the CNRS, the IRD and the itrop HPC (South Green Platform in Montpellier for providing HPC resources that have contributed to the research results reported within this study (https://bioinfo.ird.fr/).

## Author Contributions

**Conceptualization:** Christian Selinger, Samuel Alizon.

**Data curation:** Christian Selinger.

**Formal analysis:** Christian Selinger.

**Funding acquisition:** Samuel Alizon.

**Investigation:** Christian Selinger, Samuel Alizon.

**Methodology:** Christian Selinger, Samuel Alizon.

**Resources:** Samuel Alizon.

**Software:** Christian Selinger.

**Supervision:** Samuel Alizon.

**Validation:** Christian Selinger.

**Visualization:** Christian Selinger.

**Writing – original draft:** Christian Selinger, Samuel Alizon.

**Writing – review & editing:** Christian Selinger, Samuel Alizon.

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
