## [Decision Letter · Decision Letter 0]

23 Jun 2021

Dear Dr. Selinger,

Thank you very much for submitting your manuscript "Reconstructing contact network structure and cross-immunity patterns from multiple infection histories" for consideration at PLOS Computational Biology. As with all papers reviewed by the journal, your manuscript was reviewed by members of the editorial board and by several independent reviewers. The reviewers appreciated the attention to an important topic. Based on the reviews, we are likely to accept this manuscript for publication, providing that you modify the manuscript according to the review recommendations.

The reviewers and editors were excited about this innovative approach to an important question in infectious disease dynamics, and the combination of simple models with advanced computational methods. Overall the suggested edits are minor, and most of them are explicitly optional, and can be addressed - or not - at the authors discretion. The editors request that you focus on a) fixing typos/sources of confusion/missing SI figures and expanding figure captions, b) consider adding an initial diagram which explains the model and the idea behind "barcodes", c) provide slightly more context into how this theoretical work relates to real-world inference scenarios of interest for particular types of infections.

Sincerely,

Alison L. Hill

Associate Editor

PLOS Computational Biology

Virginia Pitzer

Deputy Editor-in-Chief

PLOS Computational Biology

[LINK]

Reviewer's Responses to Questions

**Comments to the Authors:**

Reviewer #1: In this study the authors show that multiple infection histories of individuals can be used to identify characteristics of the underlying transmission network structure and that they can also be used to detect immunological interference between pathogens. They do this by developing a simulator of a stochastic epidemic model consisting of multiple infections on a network and introduce the idea of ‘infection barcodes’ (taken from barcode theory in computational topology) that encodes individual infection histories in the form of times of infection onset and recovery, and the type of pathogen.

They are able to show that the similarity matrices obtained by comparing infection barcodes between pairs of individuals are correlated with network adjacency matrices and that an individual’s degree of connectivity in the network can be inferred from their connectivity in the space of the infection barcodes. Furthermore, they also show that analyzing pathogen co-occurrence patterns within hosts can be used to detect immunological interference.

Overall, I find the proof of concept described in this work to be very interesting and useful as the transmission network plays an important role in the spread of epidemics but it is generally very difficult to characterize. The methodology used in the paper is clear and I only have a few minor comments.

Introduction

• Line 18: ‘not only’ might be missing between ‘that’ and ‘the average’

• Line 27: Opening quotation mark on ‘contact’

• Line 65: Opening quotation mark on ‘age’

Methods

• Line 104: ‘Upon network initialization, we randomly seeded outbreaks of one infection per genotype, allowing for up to four genotypes per host.’ Given that the outbreaks were seeded at the same time, how do you expect the results to be affected when the seeding occurs at different points in time? Naively I would expect this to decrease the difference between the genotype specific and the genotype agnostic case.

• Line 138: ‘To better handle the computational workload of multiple infections on networks, we stored the infection histories in an interval tree data structure containing the times of onset and end of each infection episode, the genotype, and the host number’. In reality there might be uncertainties in the times of onset and end of infection episodes or the knowledge of infection episodes for individuals might be incomplete. How sensitive are the results to such uncertainties?

• Lines 149-157: Line 149 for example says, ‘In the case of recovery of host i from infection with genotype g, we wrote the end time of the infection episode to the interval tree, removed the recovered infection from the rate vector, and added potential infections with genotypes other than g from the host's network neighborhood.’ This implies that an individual can’t get reinfected by g once they have recovered. Was this written just as an example? As for instance this is not the case for all the genotypes in immunity setting I_2.

• Line 161: Opening quotation mark on ‘clustered’

Results

• Fig 1: x-axis is missing units of time

• Fig 2: I think the results of this section would become clearer if the Mantel test r-values were also added to this figure. Also, on first glance I found it a little confusing to see the correlation being positive for the adjacency matrix (in Fig.S1) but negative for the shortest path. Might help if this difference is described briefly in the Methods.

• Line 320: ‘Cross-immunity assumptions allowing for multiple infections with highly diverse infection barcodes (e.g. full cross-immunity, asymmetric decreasing) resulted in higher significance levels’. Is there a typo in this line? As it seems that the first three columns in Fig.2 (i.e. apart from ‘full cross-immunity’ and ‘asymmetric decreasing’) have higher significance levels.

• Line 322: ‘Significance levels and correlation values appeared to be higher when tested for global network properties given by the shortest path matrix (bottom), which is consistent with the barcode similarity being a quantitative measure.’ Is this statement also only restricted to the results in the first three columns of Fig.2? Also, is there an intuitive reason why the significance levels are so low (even for 100% sampling) for the shortest path full-cross immunity case?

• Line 325: ‘We found that the correlations were highly significant even at a network sampling rate of 10%’. Looking at Fig.2 this sentence comes across a bit strong, specifying the cases for which this is true will help.

Reviewer #2: Overview

Selinger & Alizon present a wonderfully well-written manuscript and simulation study to explore how multiple infection histories can be used to infer network properties and immunological interference among co-circulating pathogens. This paper was justified, with very clearly written methods, and intuitive results. I have a few “major” comments, although they are more questions out of curiosity than necessary items to implement.

Major comments

-Have the authors considered differences in susceptibility across hosts? For example, younger individual may be more susceptible to infection with influenza due to less (or no) pre-existing immunity. This could also be the case for individuals with co-morbidities or individuals that are immunocompromised. Could these types of individual-level differences be implemented in this model?

-I am wondering if the authors considered including a conceptual diagram for the barcode concept. I think this could be very useful to include in the methods, especially for those readers that are not familiar with barcode theory (probably most readers and infectious disease dynamics folks).

-Throughout the results, it would be very helpful to contextualize the findings more. Specifically, I think the authors could explain if results were intuitive or not; they do this to some degree, but I think even more would be very useful. For example, did the authors expect the partial cross-immunity scenario to most accurately predict node degree? If so why or why not?

Minor comments

-Line 27: Could you add one reference for each: definition issues and assessing the appropriateness of various types of data?

-Line 38: upon first definition, it is not immediately clear to me what “taking advantage of multiple infection as a unique descriptor of a host’s position within the network means.” Could you please clarify this?

-Line 41: although I agree that the norm is for longitudinal data to be used in more simple statistical models rather than dynamical models, there are a few very nice recent works that incorporated longitudinal data into individual-level dynamical models (Ranjeva et al. 2017- PMID: 29208707 and Ranjeva et al. 2019- doi: 10.1038/s41467-019-09652-6)

-Line 61: upon first introduction, I don’t know what barcode theory is. It would be helpful to provide a very brief definition of barcode theory upfront in the introduction.

-Fig 1. I am surprised that diversity decreases as much in the homologous setting as in the full cross-immunity setting? Is this expected? It might be nice to quantify the differences between panels in Figure 1… like are the diversity, MOI, and time-averaged fraction of nodes infected statistically significantly different across panels?

-it would be helpful in Figure 5 to include the immunity input matrix too, maybe there could be one additional panel with the 5 immunity input matrices shown, this would allow an easier comparison between co-occurrence score and the potential (or lack thereof) for homo / heterologous protection

Additional comments from the Editors:

The Author Summary is well-written but it might be a little too technical for the intended audience

<!--?xml version="1.0" encoding="UTF-8"?

**Have the authors made all data and (if applicable) computational code underlying the findings in their manuscript fully available?**

Reviewer #1: Yes

Reviewer #2: **No: **They have not yet, but they say this will upon publication.

PLOS authors have the option to publish the peer review history of their article (what does this mean?). If published, this will include your full peer review and any attached files.

Reviewer #1: No

Reviewer #2: No

Figure Files:

Data Requirements:

Reproducibility:

References:

---

## [Editor Report · Decision Letter 1]

23 Aug 2021

Dear Dr. Selinger,

We are pleased to inform you that your manuscript 'Reconstructing contact network structure and cross-immunity patterns from multiple infection histories' has been provisionally accepted for publication in PLOS Computational Biology.

Best regards,

Alison L. Hill

Associate Editor

PLOS Computational Biology

Virginia Pitzer

Deputy Editor-in-Chief

PLOS Computational Biology

---

## [Editor Report · Acceptance letter]

8 Sep 2021

PCOMPBIOL-D-21-00635R1 

Reconstructing contact network structure and cross-immunity patterns from multiple infection histories

Dear Dr Selinger,

I am pleased to inform you that your manuscript has been formally accepted for publication in PLOS Computational Biology. Your manuscript is now with our production department and you will be notified of the publication date in due course.

With kind regards,

Amy Kiss
